# ENDOWING PROTEIN LANGUAGE MODELS WITH STRUCTURAL KNOWLEDGE

## ABSTRACT

Protein language models have shown strong performance in predicting function and structure across diverse tasks. These models undergo unsupervised pretraining on vast sequence databases to generate rich protein representations, followed by finetuning with labeled data on specific downstream tasks. The recent surge in computationally predicted protein structures opens new opportunities in protein representation learning. In our study, we introduce a novel framework to enhance transformer protein language models specifically on protein structures. Drawing from recent advances in graph transformers, our approach refines the self-attention mechanisms of pretrained language transformers by integrating structural information with structure extractor modules. This refined model, termed the Protein Structure Transformer (PST), is further pretrained on a protein structure database such as AlphaFoldDB, using the same masked language modeling objective as traditional protein language models. Our empirical findings show superior performance on several benchmark datasets. Notably, PST consistently outperforms the foundation model for protein sequences, ESM-2, upon which it is built. Our code and pretrained models will be released upon publication.

## 1 INTRODUCTION

Proteins are fundamental building blocks of life, supporting an abundance of biological functions. Their functional activities range from guiding cell division, intra/inter cellular transport, to reaction catalysis and signal transduction. These multifaceted tasks are achieved because proteins, initially formed as linear polymer chains, undergo a transformation to self-assemble into 3D folds. The geometry and physico-chemical characteristics of these folds dictate a range of interactions essential for realizing a specific biological role. Consequently, disruptions in these processes can underlie numerous diseases. However, there is a gap in understanding between a protein's sequence and structure and depicting its overarching function. Bridging this gap can pave the way for breakthroughs, especially in the realm of therapeutic innovations (Ren et al., 2023; Bhardwaj et al., 2022).

As databases of protein sequences have grown exponentially over the decades, data-driven approaches for modeling proteins at scale made substantial progress. Recent studies highlight the efficacy of protein language models (PLMs) (Rives et al., 2021). By undergoing unsupervised pretraining on large databases of hundreds of millions of sequences, these models capture both secondary and tertiary protein structures within the representation space, and can be subsequently used for atomic-level structure prediction (Lin et al., 2023b). PLMs have also shown remarkable proficiency in predicting various protein functions, including the effects of mutations (Meier et al., 2021), metal ion binding (Hu et al., 2022a), and antibiotic resistance (Hu et al., 2022a).

Despite advances in sequence-based models, the potential of using protein structure for functional prediction remains largely untapped. Delving into this relationship is paramount, given the intrinsic connection between a protein's structure and its function. Historically, developing structure-aware models was challenging due to limited number of resolved protein structures. However, breakthroughs like AlphaFold (Jumper et al., 2021) have facilitated the generation of extensive, accurate protein structure databases (Varadi et al., 2022), paving the way for more comprehensive structure-based model development. Early efforts (Gligorijević et al., 2021; Wang et al., 2022) introduced geometry or graph-based models that are pre-trained on protein structure databases often using complex pre-training pipelines. However, their performance still lags behind PLMs despite drawing on,

in principle, richer input data. More recent studies (Zhang et al., 2022; 2023b) aim to enhance PLMs with protein structure data. However, many such methods simply overlay a structure encoder upon the sequence representations derived from existing PLMs like Evolutionary scale modeling (ESM) (Rives et al., 2021; Lin et al., 2023b), yielding suboptimal models in terms of parameter efficiency. Additionally, the need to finetune these models for each downstream task amplifies computational costs. An alternative research avenue involves deducing the sequence solely from the structure, often referred to as the inverse folding problem (Ingraham et al., 2019; Jing et al., 2020; Gao et al., 2022). However, these methods do not primarily aim to predict protein function and avoid using sequence data as input.

In this work, we propose a novel framework that seamlessly incorporates protein structural information into existing transformer PLMs. Drawing on the progress of graph transformers (Chen et al., 2022), we enhance the cutting-edge PLM, ESM-2 (Lin et al., 2023b) by fusing structure extractor modules within its self-attention architecture. The entire model can then be further pretrained by simply optimizing its original masked language modeling objective on a protein structure database, such as AlphaFoldDB. Notably, we highlight that *refining only the structure extractors while keeping the backbone transformer frozen can already yield substantial improvements*, addressing the concern of parameter efficiency. The refined model, termed Protein Structure Transformer (PST), can serve dual purposes: extracting refined protein representations, or finetuning for specific downstream applications. Despite the simplicity of our method, empirical results demonstrate its superiority over ESM-2 in diverse function and structure prediction tasks. Contrasting with many prior structure-based models (Zhang et al., 2022; 2023b) that require exhaustive finetuning, PST's protein representations can serve broad purposes by merely introducing a linear or multilayer perceptron atop them. Finally, our method sets a new benchmark in Enzyme Commission number and Gene Ontology term predictions (Gligorijević et al., 2021), along with multiple tasks from the new ProteinShake benchmarks[1].

## 2 RELATED WORK

Protein representation models can be categorized on their primary input data modality into three main types: sequence-based, structure-based, and hybrid models which utilize both sources of information. Within each of these categories, models can adopt either a self-supervised or an end-to-end supervised approach.

**Sequence-based models.** Transformers, introduced by Vaswani et al. (2017), demonstrated strong performance in protein function prediction when self-supervised pretrained on large protein sequence datasets (Rao et al., 2019; Hu et al., 2022a). However, Shanehsazzadeh et al. (2020) later revealed that even without pretraining, CNNs can surpass the performance of these pretrained transformers.

Subsequent models, including ProtBERT-BFD (Elnaggar et al., 2021), xTrimoPGLM (Chen et al., 2023), Ankh (Elnaggar et al., 2023), ESM-1b (Rives et al., 2021), and its evolved version, ESM-2 (Lin et al., 2023b), have further enhanced the modeling capabilities within the protein domain. This has led to strong predictive outcomes on a variety of downstream tasks, such mutation effect prediction (Meier et al., 2021), enzymatic turnover prediction (Kroll et al., 2023b), and more (Kroll et al., 2023a; Lin et al., 2023a; Hu et al., 2022b).

**Structure-based models.** Given the strong evolutionary conservation of protein structures and their direct influence on protein functionality Illergård et al. (2009), structure-based models often provide more accurate predictions Townshend et al. (2020). Several techniques have been proposed, ranging from 3D Convolutional Neural Networks (CNNs) (Derevyanko et al., 2018), point cloud and voxel models (Yan et al., 2022; Mohamadi et al., 2022), to graph-based representations.

Graph Neural Networks (GNNs), employed in models such as GVP (Jing et al., 2020) and IEConv (Hermosilla et al., 2021), offer inherent flexibility to encode protein-specific features as node or edge attributes. Furthermore, with the recent proliferation of protein folding models (Jumper et al., 2021), structure-based models can leverage abundant protein structure datasets, although sequence datasets still remain predominant.

---

[1] https://proteinshake.ai

**Hybrid models.** Hybrid models rely both on sequence and structural information to offer enriched protein representations. For instance, DeepFRI by Gligorijević et al. (2021) combined LSTM-based sequence extraction with a graph representation of protein structures. Similarly, Wang et al. (2022) utilized ProtBERT-BFD embeddings as input features for the GVP model, resulting in enhanced predictive capabilities.

A recent approach, ESM-GearNet, proposed by Zhang et al. (2023b), explored different ways of combining ESM-1b representations with GearNet. The model set new benchmarks across various function prediction tasks, though requiring fine-tuning separately on each task. In parallel, the work of Zheng et al. (2023) also delved into similar ideas with a particular focus on the inverse protein folding task. Unlike these prior works that simply append a structure-aware adapter to a PLM model, our approach incorporates structural information directly into *each* self-attention layer of the PLM. This strategy allows for a deeper interaction between structural and sequence features, setting our model apart from previous models.

## 3 METHODS

We first introduce the state-of-the-art sequence models, ESM-2. Then we explain how to represent proteins as graphs and how to adapt the ESM-2 models to account for structural information.

### 3.1 BACKGROUND: EVOLUTIONARY SCALE MODELING

ESM is a family of transformer protein language models. The initial version, called ESM-1b (Rives et al., 2021), was trained with 650M parameters and 33 layers on a high-diversity sparse dataset featuring UniRef50 representative sequences. The authors have recently unveiled an advanced generation, ESM-2 (Lin et al., 2023b), which ranges from 8M to 15B parameters. ESM-2 brings architectural enhancements, refined training parameters, and expands both computational resources and size of the input data. When compared on equal parameter grounds, the ESM-2 model consistently outperforms its predecessor, ESM-1b.

The ESM-2 language model employs the masked language modeling objective. This is achieved by minimizing

$$\mathcal{L}_{\text{MLM}} = \mathbb{E}_{x \sim X} \mathbb{E}_M \sum_{i \in M} - \log p(x_i \,|\, x_{/M}), \tag{1}$$

for any given protein sequence $x$, a random set of indices, $M$, is selected to undergo masking. This entails substituting the actual amino acid with a designated mask token. Subsequently, the log likelihood of the genuine amino acid type is maximized, considering the unmasked amino acids $x_{/M}$ as the context.

For the training phase, sequences are uniformly sampled across approximately 43 million UniRef50 training clusters, derived from around 138 million UniRef90 sequences. This ensures that during its training, the model is exposed to roughly 65 million distinct sequences.

#### 3.1.1 ESM-2 MODEL ARCHITECTURE

The ESM-2 models use BERT-style encoder-only transformer architecture with certain modifications (Vaswani et al., 2017; Kenton & Toutanova, 2019). These models are constructed with multiple stacked layers, each comprising two primary building blocks: a self-attention layer followed by a feed-forward layer. For the self-attention mechanism, token features $X \in \mathbb{R}^{n \times d}$ are first projected to the query ($Q$), key ($K$) and value ($V$) matrices through linear transformations, as given by:

$$Q = XW_Q, \quad K = XW_K, \quad V = XW_V, \tag{2}$$

where $W_* \in \mathbb{R}^{d \times d_{\text{out}}}$ represent trainable parameters. The resulting self-attention is defined as

$$\text{Attn}(X) = \text{softmax}(\frac{QK^\top}{\sqrt{d_{\text{out}}}})V \in \mathbb{R}^{n \times d_{\text{out}}}. \tag{3}$$

It worth noting that multi-head attention, which concatenates multiple instances of Eq. 3, is commonly adopted and has been empirically effective (Vaswani et al., 2017). Then, the output of the

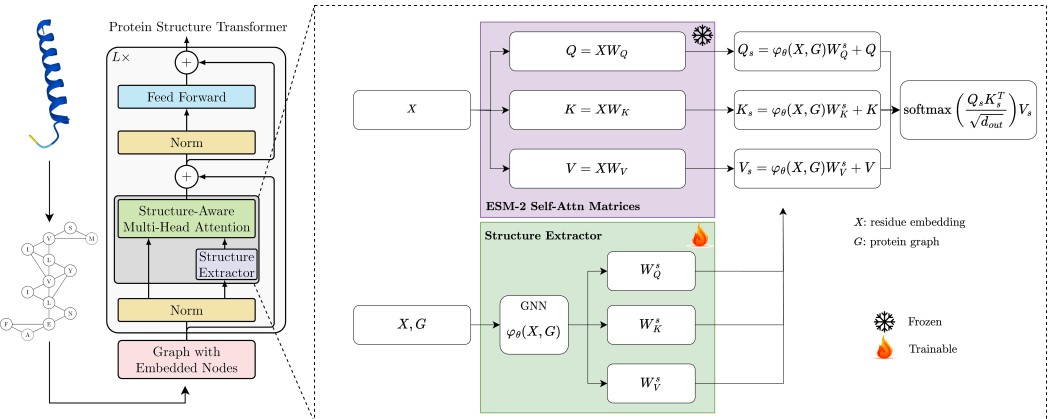

Figure 1: Overview of the proposed Protein Structure Transformer. A protein 3D structure is converted to an ordered graph with nodes representing amino acid types and edges linking any pair of residues within a specified threshold (8Å). Then, we sample a fraction of nodes and mask them using a special mask token. The output of Protein Structure Transformer (PST) is fed into a linear prediction head to predict the original amino acid types. Compared to ESM-2, PST uses a GNN to extract local structural features around each node before computing the self-attention at each transformer layer. The PST model is initialized with the pretrained ESM-2 weights.

self-attention is followed by a skip-connection and a feedforward network (FFN), which jointly compose a transformer layer, as shown below:

$$
\begin{aligned}
X' &= X + \text{Attn}(X), \\
X'' &= X' + \text{FFN}(X') := X' + \text{ReLU}(X'W_1)W_2,
\end{aligned}
\tag{4}
$$

where $W_1$ and $W_2$ are trainable parameters and $\text{ReLU}$ denotes the rectifier activation function. While the original transformer uses absolute sinusoidal positional encoding to inform the model about token positions, ESM-2 leverages the Rotary Position Embedding (RoPE) (Ho et al., 2019), enabling the model to extrapolate beyond its trained context window. The main goal of our PST is to slightly modify ESM-2 such that it can take protein structural information into account.

## 3.2 PROTEIN STRUCTURE TRANSFORMER

Recent advances in graph representation learning facilitated the adaptation of transformer architectures to process graph-structured data, leading to the emergence of what are formally termed "graph transformers". In particular, structure-aware transformers (Chen et al., 2022) meld the vanilla transformer framework with graph neural networks. This integration proficiently captures complex interactions inherent to local structures, offering substantial improvement over conventional GNNs. Considering the intrinsic ability to represent protein structures as graphs, these advances position graph transformers as particularly adequate for modeling protein structures. In the following, we present the methodology for representing a protein as a graph and delineate the modifications implemented in the ESM-2 model to construct our dedicated protein structure transformer. The model architecture and the complete pretraining process are illustrated in Figure 1 and Figure 5, respectively. In summary, we employ a structure extractor, *e.g.,* a shallow GNN, that modifies *each self-attention calculation* within the ESM-2 model. Note that the structure extractor may vary across layers. It takes the residue embedding from the specific layer it is applied to, along with structural information, and produces an updated residue embedding. This updated embedding is then transformed to update the query, key, and value matrices provided by the ESM-2 for the self-attention mechanism.

**Protein graph representation.** Proteins can be conceptualized as ordered graphs, denoted as $G$. In this representation, the node order encodes the sequence annotated with amino acid types. A connection is drawn between two nodes when the spatial distance between the corresponding residues remains within a specified threshold. Based on our empirical studies, this threshold is set at 8.0 Angstroms, primarily because a majority of local intermolecular interactions manifest within this

range (Bissantz et al., 2010). The residue distances might serve as potential edge attributes for the graph; however, omitting distance information proved more effective, as discussed in Section 4.4.

**Protein Structure Transformer construction.** The PST, built upon an ESM-2 model, accepts the protein graph as input. A distinguishing feature is the substitution of traditional self-attention mechanisms with structure-aware attention mechanisms (Chen et al., 2022). Concretely, the residue embeddings at each self-attention layer, in conjunction with the graph structure, are processed through a structure extractor $\varphi_\theta(X, G) \in \mathbb{R}^{n \times d}$ to ascertain local structural contexts surrounding each residue, prior to computing the self-attention. These deduced structural embeddings subsequently undergo a linear transformation prior to be added to the query, key and value matrices, as expressed by:

$$Q_s = Q + \varphi_\theta(X, G)W_Q^s, \quad K_s = K + \varphi_\theta(X, G)W_K^s, \quad V_s = V + \varphi_\theta(X, G)W_V^s. \quad (5)$$

Here, $W_*^s \in \mathbb{R}^{d \times d_{\text{out}}}$ signifies trainable parameters initialized at zero, ensuring that the nascent model, prior to any training, mirrors the base ESM-2 model. Pertinently, the structure extractor can be any function operating on the subgraphs around each node. For computational reasons, we select a commonly utilized graph neural network, specifically GIN (Xu et al., 2018).

**Pretraining the PST.** The PST, initialized with the pretrained weights of an ESM-2 model, is further pretrained on a protein structure database, employing the masked language modeling objective same as the base ESM-2 model. One can opt to either update solely the parameters within the structure extractors $\theta$ and $W_s$ or refine the entire model. Comparative evaluations of these strategies will be discussed in the Section 4.4.

## 4 EXPERIMENTS

In this section, we conduct a comprehensive evaluation of PST models over a wide range of tasks encompassing both structure and function predictions, sourced from various datasets. A focal point of our experiments is to show the versatility of PST representations across relevant downstream tasks, even in the absence of any fine-tuning. Additionally, we undertake an in-depth analysis of the model's architecture, the employed training strategy, and the degree of structural information needed, aiming to identify the main contributors to performance. We summarize the results of our experiments as follows:

- PST achieves state-of-the-art performance on function prediction tasks including enzyme and gene ontology classification, as well as fold classification.

- The inherent adaptability of PST's representations is evidenced by their robust performance across a variety of tasks, *without the need for task-specific fine-tuning on the representations*.

- In a comparative analysis with the state-of-the-art sequence model ESM-2, PST consistently exhibits superior performance, emphasizing the value of incorporating structural knowledge. Interestingly, this enhancement is more pronounced for smaller ESM-2 models, suggesting a heightened impact of structural data on them.

- Incorporating more subtle structural data into PST augments its pretraining efficacy. However, this augmentation may inadvertently compromise the model's performance in downstream applications. Such a phenomenon is potentially attributable to the inherent simplicity of the masked language modeling objective, suggesting a possible need for more intricate objectives when integrating advanced structural data.

- While a full pretraining of the PST model yields optimal results, a targeted pretraining of just the structure extractors produces both sequence and structural representations within a unified model framework. This approach is more parameter efficient while retaining efficacy.

## 4.1 EXPERIMENTAL SETUP

### 4.1.1 DATASETS

**Function and structure prediction datasets.** To evaluate the performance of the PST models and compare them to a number of comparison partners, we use several sets of benchmarks covering a diverse range of tasks. We use a set of experimentally resolved protein structures and predict their function. The function of a protein is either encoded as their Gene Ontology (GO) term or Enzyme Commission (EC) number. The curation and splitting of these datasets are detailed in Gligorijević et al. (2021). Each of those downstream tasks consists of multiple binary classification tasks. For each task, we calculate the $F_{max}$ score as well as the AUPR score for evaluation. Additionally, we consider the fold classification dataset curated by Hou et al. (2018). More information on those datasets and metrics is provided in Appendix C.5 and C.2.

**ProteinShake benchmark datasets** We use ProteinShake, a convenient library making the evaluation of machine learning models across a wide variety of biological tasks easier. The tasks in this library include functional and structural classification tasks such as enzyme commission and gene ontology class predictions (similar to the previous benchmark, but on a different set of proteins), a task where one seeks to classify the protein family (Pfam) label of a protein (Mistry et al., 2021), as well as a classification task of the SCOPe labels of proteins (Chandonia et al., 2022). Another task is a residue-level classification task identifying the residues involved in a binding pocket of a protein (Gallo Cassarino et al., 2014). Importantly, the software library provides metrics and precomputed splits on the basis of sequence similarity and structural similarity for each task. In this work, we exclusively use the most stringent structure-based splits provided by ProteinShake. The documentation and implementation of this library can be found at `https://proteinshake.ai`.

**Variant effect prediction (VEP).** An common use of protein representations is to predict the effect of a mutation at a given position in the sequence, saving valuable bench time (Riesselman et al., 2018; Meier et al., 2021). Here, we use the computationally predicted AlphaFold structure of each wild-type sequence as input to the structural model, and swap each mutated position in the resulting graph before computing the (mutated) protein's representation. The collection of 38 proteins evaluated here were first presented in (Riesselman et al., 2018) and further used in (Meier et al., 2021). The Spearman's correlation coefficient $\rho$ between a model's predicted scores and an experimentally acquired outcomes is then used to benchmark models. Note that all predictions are made *without any fine-tuning*, further allowing us to fairly compare the quality of the representations obtained from PST compared to ESM-2. The performance of our model on each protein, as well as more information on this benchmark can be found in Appendix C.6.

### 4.1.2 TRAINING

**Pretraining.** We build PST models on several ESM-2 variants, including `esm2_t6_8M_UR50D`, `esm2_t12_35M_UR50D`, `esm2_t30_150M_UR50D`, `esm2_t33_650M_UR50D`. The two largest models did not fit in the VRAM of our GPUs. For each ESM-2 model, we endow it with structural knowledge by incorporating the structure extractor in every self-attention, as described in Section 3.2. We use GIN (Xu et al., 2018) as our structure extractor with two GNN layers, as suggested in Chen et al. (2022). We pretrain all four PST models on AlphaFold's SwissProt subset of 542,378 predicted structures (Jumper et al., 2021; Varadi et al., 2022). We initialize the model weights with the corresponding pretrained ESM-2 weights, except for the structure extractors which are initialized randomly (for $\theta$) or to zeroes for the linear projection parameters $W_*^s$.

**Task-specific models.** In order to evaluate the generalizability of the protein representations from PST models, we choose to fix the representations instead of fine-tuning them for each specific task. This is meaningful in practice, as it can save a lot of computation. Specifically, we compute the average representation across residues in the protein and concatenate the representations from each layer. Once the representations are extracted, we train a task-specific classifier (classification head) atop them. We choose an MLP for multilabel classification and a linear model for other types of classification. More details can be found in Appendix C.4.

Table 1: Performance of PST models compared to others on protein function prediction tasks. MVC: multiview contrast. Relevant comparison partners have been selected from Zhang et al. (2023b).

| Method | EC | | GO-BP | | GO-MF | | GO-CC | | Fold Class (ACC) | | |
|---|---|---|---|---|---|---|---|---|---|---|---|
| | $F_{max}$ | AUPR | $F_{max}$ | AUPR | $F_{max}$ | AUPR | $F_{max}$ | AUPR | Fold | Super. | Fam. |
| *End-to-end training* | | | | | | | | | | | |
| CNN | 0.545 | 0.526 | 0.244 | 0.159 | 0.354 | 0.351 | 0.287 | 0.204 | 11.3 | 13.4 | 53.4 |
| Transformer | 0.238 | 0.218 | 0.264 | 0.156 | 0.211 | 0.177 | 0.405 | 0.210 | 9.22 | 8.81 | 40.4 |
| DeepFRI | 0.631 | 0.547 | 0.399 | 0.282 | 0.465 | 0.462 | 0.460 | 0.363 | 15.3 | 20.6 | 73.2 |
| ESM-1b | 0.864 | 0.889 | 0.452 | 0.332 | 0.657 | 0.639 | 0.477 | 0.324 | 26.8 | 60.1 | 97.8 |
| ProtBERT-BFD | 0.838 | 0.859 | 0.279 | 0.188 | 0.456 | 0.464 | 0.408 | 0.234 | 26.6 | 55.8 | 97.6 |
| LM-GVP | 0.664 | 0.710 | 0.417 | 0.302 | 0.545 | 0.580 | **0.527** | **0.423** | – | – | – |
| GearNet MVC | 0.874 | 0.892 | 0.490 | 0.292 | 0.654 | 0.596 | 0.488 | 0.336 | **54.1** | 80.5 | **99.9** |
| ESM-1b-Gearnet MVC | 0.894 | 0.907 | **0.516** | 0.301 | **0.684** | 0.621 | 0.506 | 0.359 | – | – | – |
| ESM-2-Gearnet MVC | 0.896 | – | 0.514 | – | 0.683 | – | 0.497 | – | – | – | – |
| ESM-2 (finetuned) | 0.893 | 0.901 | 0.460 | 0.308 | 0.663 | 0.627 | 0.427 | 0.331 | 38.5 | 81.5 | 99.2 |
| PST (finetuned) | **0.897** | **0.919** | 0.489 | **0.348** | 0.675 | **0.648** | 0.475 | 0.375 | 42.3 | **85.9** | 99.7 |
| *Fixed representations + classification head* | | | | | | | | | | | |
| Ankh | 0.870 | 0.897 | 0.466 | 0.347 | 0.650 | **0.647** | 0.500 | 0.393 | 33.3 | 74.9 | 98.7 |
| Gearnet MVC | 0.826 | 0.852 | 0.428 | 0.303 | 0.594 | 0.589 | 0.433 | 0.337 | 38.5 | 68.3 | 98.8 |
| ESM-2 | 0.892 | 0.910 | 0.509 | 0.355 | **0.686** | 0.629 | 0.529 | **0.394** | 39.7 | 80.4 | 98.8 |
| PST | **0.899** | **0.918** | **0.513** | **0.371** | **0.686** | 0.637 | **0.541** | 0.387 | **40.9** | **83.6** | **99.4** |

## 4.2 COMPARISON TO STATE-OF-THE-ART METHODS FOR PROTEIN FUNCTION AND STRUCTURE PREDICTION

In this section, we evaluate the performance of PST models against several state-of-the-art counterparts on several function and structure prediction datasets, as shown in Table 1. The comparison partners include sequence models without pretraining such as CNN (Shanehsazzadeh et al., 2020), Transformer (Rao et al., 2019), with pretraining such as ProtBERT-BFD (Elnaggar et al., 2021), Ankh (Elnaggar et al., 2023), ESM-1b (Rives et al., 2021), ESM-2 (Lin et al., 2023b), structure models with pretraining such as DeepFRI (Gligorijević et al., 2021), LM-GVP (Wang et al., 2022), GearNet MVC (Zhang et al., 2022), and a hybrid model ESM-GearNet MVC (Zhang et al., 2023b) which integrates ESM-1b and GearNet MVC, as well as its newer versions ESM-2-Gearnet MVC and ESM-2-Gearnet SiamDiff, recently introduced in Zhang et al. (2023a).

While previous studies focusing on training independent models for each individual, we take a distinct approach, aiming to assess the universality of protein representations from pretrained models. To this end, we fix the representations across tasks following the procedure described in Section 4.1.2. This procedure is equally applied to both GearNet MVC and ESM-2. Contrary to the claims in the work by Zhang et al. (2022), our analysis suggests that ESM outperforms GearNet MVC in the sense that it generates more general-purpose representations. Notably, our PST models surpass ESM-2 in performance, particularly evident in the fold classification task where PST models outperform ESM-2 significantly, underscoring the efficacy of structure-centric models in discerning protein structural variations. Furthermore, PST models with fixed representations demonstrate competitive or superior performance against several end-to-end models while demanding substantially reduced computational time, emphasizing their enhanced adaptability and applicability in diverse real-world scenarios.

Additionally, we engage in task-specific fine-tuning of our PST models. Though computationally more intensive than employing fixed representations, the fine-tuned PST models consistently achieve superior AUPR scores across function prediction tasks. It is worth noting that PST surpasses the state-of-the-art ESM-GearNet MVC, a model that integrates ESM in a decoupled fashion.

## 4.3 STRUCTURE VS. SEQUENCE: A COMPARATIVE ANALYSIS OF PST AND ESM-2 MODELS

In this section, we provide further evidence that our PST model consistently outperforms the ESM-2 model on which is based. As highlighted in Section 4.2, across a wide range of function preditction tasks, employing PST embeddings alongside an MLP classification head consistently achieves enhanced performance, relative to using ESM embeddings with a comparable MLP head.

To mitigate the potential influences of optimization and inherent randomness in training the MLP, we opt for a simple linear model over the representations for a variety of tasks from ProteinShake. As

Table 2: Comparison of PST and ESM-2 on ProteinShake tasks and VEP datasets. Details of evaluation metrics can be found in Appendix C.2.

| Method | GO $F_{max}$ | EC ACC | Protein Family ACC | Binding Site MCC | Structural Class ACC | Zero-shot VEP mean $|\rho|$ |
|---|---|---|---|---|---|---|
| ESM-2 | 0.648 | 0.858 | 0.698 | 0.431 | 0.791 | 0.489 |
| PST | 0.650 | 0.883 | 0.704 | 0.436 | 0.797 | 0.501 |

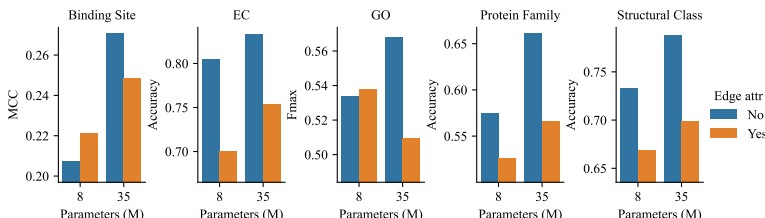

Figure 2: Performance of PST models trained with and without distance information serving as edge attributes.

seen in Table 2, PST exhibits consistent superiority over ESM-2 across these tasks. This distinction is particularly pronounced in the EC classification, where PST markedly surpasses ESM-2.

Additionally, we compare PST with ESM-2 in the context of zero-shot variant prediction tasks, conducted *without any further tuning*. PST equally outperforms ESM-2 in terms of average spearman's rank correlations. The better results in tasks like binding site detection and VEP imply that PST not only offers enhanced protein-level representations but also refines residue-level representations.

## 4.4 HYPERPARAMETER STUDIES

While Section 4.3 showcases the added value of PST relative to its base ESM-2 model, we now dissect the components of PST to identify what drives the performance.

**Amount of structural information required for refining ESM-2.** We start the analysis by investigating the extent of structural information required for refining the ESM-2 models. For this purpose, we construct two $\epsilon$-neighborhood graphs from protein 3D structures: one that does not incorporate distance information as edge attributes, and another enriched with 16-dimensional features related to residue distances serving as edge features. Structure extractors are then adapted accordingly to account for edge attributes. Details about these features can be found in the Appendix C.7. Subsequently, PST models, equipped with structure extractors either with or without edge attributes are pretrained. Their generated representations are then assessed using the ProteinShake datasets.

While incorporating more granular structural information augments pretraining accuracy (*e.g.,* from 47% to 55% for the 6-layer models), it brings a negative transfer to downstream tasks, as depicted in Figure 2. We note that the PST model leveraging edge attributes underperforms compared to its counterpart without edge attributes across all the tasks. Such a phenomenon could plausibly stem from the inherent simplicity of the masked language modeling objective, suggesting a potential necessity to devise more nuanced objectives when integrating advanced structural features.

**Effect of model size.** We evaluate PST models across various model sizes and measure the performance uplift relative to its base ESM-2 model. We pretrained four distinct PST models, each based on the ESM-2 models with sizes spanning 8M, 35M, 150M, 650M parameters. Owing to our employment of shallow, lightweight GINs as structure extractors, the resulting PST models maintain a parameter count that is less than double that of their base ESM-2 models.

Figure 3 presents the outcomes of our assessment. Notably, as model size increases, both ESM-2 and PST display enhanced performance across the majority of tasks, with exceptions observed in EC (ProteinShake) and Protein Family classification. While PST typically surpasses its base ESM-2 counterpart at similar model sizes, this performance gain tapers off with increasing model size. This trend is potentially due to large language models like ESM-2 being optimized for predicting atomic-

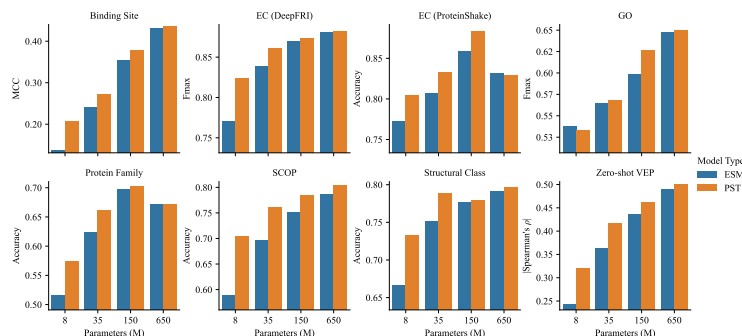

Figure 3: Performance of PST and ESM-2 across varied model sizes on ProteinShake datasets.

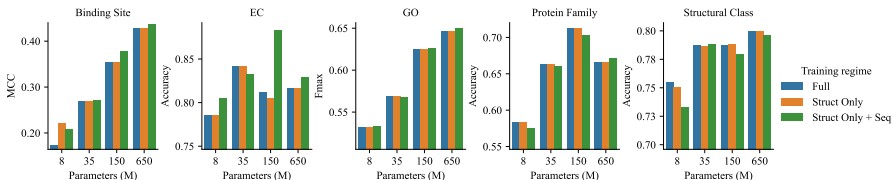

Figure 4: Effect of pretraining strategies on model performance. "Full" refers to the strategy where one updates the full model during pretraining, including both ESM-2 and structure extractor weights. "Struct Only" refers to the strategy where only the structure extractor weights are being updated during training. "Struct Only + Seq" is an extension of "Struct Only" *at inference*. By bypassing the structure extractors, the PST model is capable to obtain same sequence representations as the base ESM-2 model. Averaging both structure and sequence representations leads to "Struct Only + Seq".

level protein structures, as referenced in Lin et al. (2023b). Consequently, for scenarios constrained by computational resources, opting for structure-aware models might offer a strategic advantage.

**Pretraining strategies.** In Section 3.2, we delineate the distinctions between PST and ESM-2 models, pinpointing the addition of structure extractors as the only difference. Here, our experiments seek to ascertain if solely updating the structure extractors (including the linear transformation $W_s$) yields comparable results to a full model pretraining. Figure 4 offers a performance comparison on ProteinShake tasks, where the orange bars signify updates to the structure extractors, and the blue bars represent full model updates.

Remarkably, pretraining restricted to the structure extractors produces performance outcomes akin to a full model pretraining. Beyond parameter efficiency, this selective updating confers an additional advantage: the capability to derive the base ESM-2 model's sequence representation from the same model at inference, achieved through bypassing the structure extractors. By averaging both structure and sequence representations, we obtain enhanced representations beneficial for multiple tasks, as depicted by the "Struct Only + Seq" (green bars) in Figure 4.

## 5 DISCUSSION

In this work, we presented the PST model, a novel approach that endows pretrained PLMs with structural knowledge. Unlike previous models that need training from scratch, our approach refines existing transformer-based PLMs, amplifying their accumulated sequence knowledge.

Our evaluations reveal that PST generates general-purpose protein representations, excelling across a broad range of function prediction tasks. Notably, PST surpasses the benchmarks set by the cutting-edge PLM, ESM-2 and achieves state-of-the-art performance in various protein property prediction tasks. Finally, we envision that using more structural data and advanced pretraining objectives beyond traditional masked language modeling will unlock the full potential of larger models within the PST paradigm.

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

# Appendix

## A  LIMITATIONS AND IMPACT

This work is dedicated to deriving general-purpose protein representations through unsupervised pretraining on vast amounts of unlabeled protein structures. By incorporating structural data, our methodology outperforms sequence-based models, offering richer protein representations. This enhanced capability paves the way for deeper insights into protein research, with potential implications for areas such as protein function prediction and protein design.

**Limitations.**  The scope of this paper is limited to pretraining on 542K protein structures. With the AlphaFold Protein Structure Database now hosting over 200 million proteins, there is ample opportunity to develop more sophisticated structure-based models using expansive datasets in subsequent studies. However, as datasets grow, the challenge of scalability arises. Furthermore, our evaluations are restricted to function and structure prediction tasks. Expanding the model's application to other domains, like protein-protein interaction modeling or the design of ligand molecules guided by protein structures, holds promising potential, given their relevance in many biological applications.

**Potential negative impact.**  The power of advanced pretrained models can inadvertently be leveraged for detrimental purposes, such as crafting harmful drugs. We hope that subsequent research will address and mitigate such risks.

## B  PIPELINE OVERVIEW

An overview of the pipeline described in this work is provided in Figure 5.

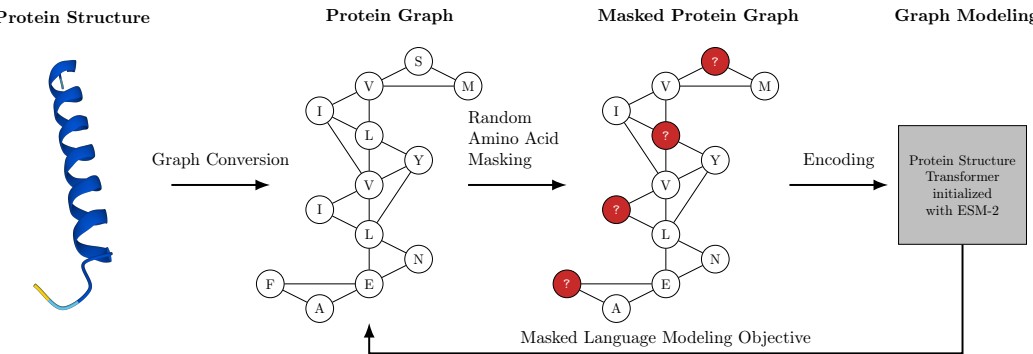

Figure 5: Overview of the pretraining pipeline of the propose Protein Structure Transformer.

## C  EXPERIMENTAL DETAILS AND ADDITIONAL RESULTS

### C.1  COMPUTATION DETAILS

We perform all the pretraining of PST models on 4 H100 GPUs. 20GiB H100 MIGs were used to fine-tune the classification heads.

### C.2  EVALUATION METRICS

In this Section, we briefly describe the performance metrics we used in the experimental evaluation of the models.

**Protein-centric F$_{\text{max}}$ score** The F$_{\text{max}}$ score is computed as follows. Let $P_x(\tau)$ be the positive functions for the protein $x$, i.e. the

Precision ($pr_i$) and recall ($rc_i$) for a given protein $i$ at threshold $\tau$ are defined as

$$pr_i(\tau) = \frac{\sum_f \mathbb{1}\left(f \in P_i(\tau) \cap T_i\right)}{\sum_f \mathbb{1}\left(f \in P_i(\tau)\right)},$$

$$rc_i(\tau) = \frac{\sum_f \mathbb{1}\left(f \in P_i(\tau) \cap T_i\right)}{\sum_f \mathbb{1}\left(f \in T_i\right)},$$

where $P_i(\tau)$ is the set of terms that have predicted scores greater than or equal to $\tau$ for a protein $i$ and $T_i$ denotes the ground-truth set of terms for that protein. Then the set $P_i(\tau) \cap T_i$ is the set of true positive terms for protein $i$.

The average precision and average recall are then defined as

$$pr(\tau) = \frac{1}{n(\tau)} \sum_{i=1}^{n(\tau)} pr_i(\tau),$$

$$rc(\tau) = \frac{1}{N} \sum_{i=1}^{N} rc_i(\tau),$$

where $N$ is the number of proteins and $n(\tau)$ is the number of proteins with at least one predicted score greater than or equal to $\tau$.

Then, we define the F$_{\text{max}}$ score as

$$\text{F}_{\text{max}} = \max_{\tau} \left\{ \frac{2 \cdot pr(\tau) \cdot rc(\tau)}{pr(\tau) + rc(\tau)} \right\}.$$

**Pair-centric AUPR** The AUPR score we use in the evaluation, following Zhang et al. (2022) is computed by area under the precision-recall curve, is defined as the average, over all protein-term pairs $(i, f)$, of the AUPR score of the classifier on that binary classification task.

**MCC** Matthew's correlation coefficient, abbreviated MCC, is defined, for a binary classification task, as follows. Let $TP$, $FP$, $TN$ and $FN$ be the number of true positives, false positives, true negatives and false negatives, respectively. Then we define

$$\text{MCC} = \frac{TP \cdot TN - FP \cdot FN}{\sqrt{(TP + FP)(TP + TN)(TN + FN)(FP + FN)}}.$$

A score of $1$ indicates perfect agreement between prediction and labels, a score of $0$ indicates no correlation among the two, and a score of $-1$ indicates perfect disagreement.

**Spearman's rank correlation coefficients $\rho$** . Spearman's rank correlation coefficient can be calculated using:

$$\rho_S = 1 - \frac{6 \sum d_i^2}{n(n^2 - 1)}$$

where $d_i = \text{R}(X) - \text{R}(Y)$ is the difference between two ranks of each observation. $\rho_S \approx 1$ or $\rho_S \approx -1$ indicates that the ranks of the two observations have a strong statistical dependence, $\rho_S \approx 0$ indicates no statistical dependence between the ranks of two observations. In this work, we look at $|\rho_S|$, as 1 and -1 carry the same meaning in the context of zero-shot VEP. See Appendix C.6 for more details.

## C.3 DETAILS ON BASELINES

We took performance values from Zhang et al. (2022) for the following baselines: CNN, Transformer, DeepFRI, ESM-1b, ProtBERT-BFD, LM-GVP, GearNet MVC (end-to-end training), ESM-Gearnet MVC. Moreover, we finetuned ourselves ESM-2 on the tasks at hand. Finally, we used

the embeddings obtained from the Ankh, GearNet MVC and ESM-2 models as input to a MLP classification head.

Table 3 reports the number of parameters for the models that we trained ourselves. We were not able to find the number of parameters for the other baselines.

Table 3: Number of parameters of the baseline models.

| Model name | Number of parameters |
|---|---|
| GearNet MVC | 20 M |
| Ankh | 1151 M |
| ESM-2 (`esm2_t33_650M_UR50D`) | 651 M |
| PST (`esm2_t33_650M_UR50D`) | 1137 M (where 486 M was trainable) |

### C.4 TRAINING DETAILS AND HYPERPARAMETER CHOICE

For the pretraining of PST models, we use an AdamW optimizer with linear Warmup and inverse square root decay, as used by Vaswani et al. (2017); Rives et al. (2021). The other hyperparameters for optimization are provided in Table 4. The number of epochs are selected such that the training time for each model is about 10 hours on 4 H100 GPUs.

After pretraining, we extract and fix representations of proteins. Once the representaions are extracted, we train a task-specific classifier. These classifiers are chosen to be an MLP for multilabel classification and a linear model for other types of classification. The MLPs consist of three layers, with the dimension of each hidden layer reduced by the factor of 2 each time. We also use a dropout layer after each activation, with its dropout ratio to be 0 or 0.5, optimized on the validation set. Each model is trained for 100 epochs using the binary cross-entropy loss, the AdamW optimizer, and the learning rate was reduced by a factor of 0.5 if the validation metric of interest is not increasing after 5 epochs.

Table 4: Hyperparameters used for pretraining PST models.

| Model name | Learning rate | Batch size | Epochs | Warmup epochs |
|---|---|---|---|---|
| `esm2_t6_8M_UR50D` | 0.0003 | 128 | 50 | 5 |
| `esm2_t12_35M_UR50D` | 0.0001 | 64 | 20 | 5 |
| `esm2_t30_150M_UR50D` | 5e-5 | 16 | 10 | 1 |
| `esm2_t33_650M_UR50D` | 3e-5 | 12 | 5 | 1 |

### C.5 DETAILS OF FUNCTION PREDICTION AND STRUCTURAL CLASSIFICATION DATASETS

**Function prediction** Similar to (Gligorijević et al., 2021) and (Zhang et al., 2022), the EC numbers are selected from the third and fourth levels of the EC tree, since the EC classification numbers is hierarchical. This forms a set of 538 binary classification tasks, which we treated here as a multi-label classification task. GO terms with training samples ranging from 50 to 5000 training samples are selected. The sequence identity threshold is also set to 95% between the training and test set, which is again in line with Gligorijević et al. (2021); Zhang et al. (2022) and Wang et al. (2022). Note that Zhang et al. (2022) also performs additional quality control steps in their code, and to ensure a fair comparison we extracted the graphs and sequence directly from their data loader.

**Fold classification** Like (Zhang et al., 2022), we follow (Hou et al., 2018) to predict the SCOPe label of a protein. There are three different splits that we use for predicting the label: *Fold*, where proteins from the same superfamily are unseen during training; *Superfamily*, in which proteins from the same family are not present during training; and *Family*, where proteins from the same family are present during training.

Table 5: Dataset statistics for downstream tasks.

| Dataset | Train | Validation | Test Fold / Superfamily / Family |
|---|---|---|---|
| **Enzyme Commission** | 15,550 | 1,729 | 1,919 |
| **Gene Ontology** | 29,898 | 3,322 | 3,415 |
| **Fold Classification** | 12,312 | 736 | 718 / 1,254 / 1,272 |

## C.6 VARIANT EFFECT PREDICTION DATASETS

A set of deep mutational scans (DMS) from diverse proteins curated by Riesselman et al. (2018) are used to benchmark sequence-based models and PST, since it was used to evaluate state-of-the-art models (Meier et al., 2021). DMS experiments are routinely used in biology to assess the sequence-function landscape of a protein by mutating one or more of its residues and evaluating the resulting mutant function with respect to the wild type protein. The function of a protein can be measured using various mechanisms, such as growth of the cell in which the (mutated) protein is expressed, resistance to antibiotics, or fluorescence, depending on the protein and experimental conditions (Fowler & Fields, 2014). In this collection of DMS experiments, a wide variety of readouts and proteins are evaluated depending on the task, and they are summarized in Supplementary Table 1 in Riesselman et al. (2018). Let $G$ be the graph representation of the wild type protein structure, $x^{\text{mut}}$ and $x^{\text{wt}}$ be the mutant and wild type sequences, respectively. Let $x_{/M}$ be a sequence with masked node indices $M$, and $M$ the total set of mutations applied to a single wild type protein. Then, the log masked marginal probability is defined as:

$$\sum_{i \in M} \log p(x_i = x_i^{\text{mut}}|x_{/M}, G) - \log p(x_i = x_i^{\text{wt}}|x_{/M}, G). \tag{6}$$

This procedure allows us to perform mutational effect prediction without any additional supervision or experimental data on each task, deriving knowledge solely from the semi-supervised pre-training process. [2] We then compute the Spearman correlation coefficient between this set of scalar values obtained for each mutation and the observed effect in the variable of interest (depending on the protein).

Figure 6 shows the performance of ESM-2 and PST representations on deep mutational scan datasets collected by Riesselman et al. (2018). The standard deviations are obtained using 20 bootstrapped samples, in accordance with previous work (Meier et al., 2021). The mean score between the ESM-2 score and PST were used for obtaining the PST model.

## C.7 EDGE ATTRIBUTES DETAILS

We follow (Ingraham et al., 2019) and extract 16 Gaussian RBFs isotropically spaced from 0 to 20 Å. This maximizes the value of the RBF kernel when the interatomic distance is close to the value of the center in the RBF kernel.

## D ILLUSTRATION OF DIFFERENT PRETRAINING STRATEGIES

The different pretraining strategies studied in section 4.4 are illustrated in Figure 7.

---

[2]We substitute the amino acid label in the wild-type structure, as AlphaFold2 authors explicitely said that point mutant structures are unreliable. See (Jumper et al., 2021) for more details.

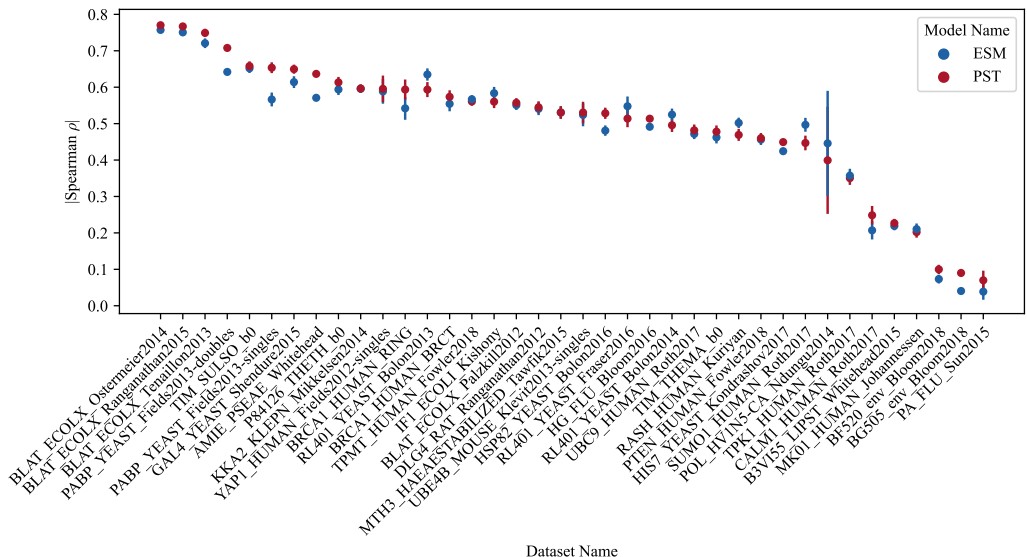

Figure 6: 33-layer ESM and PST model performance on deep mutational scanning tasks.

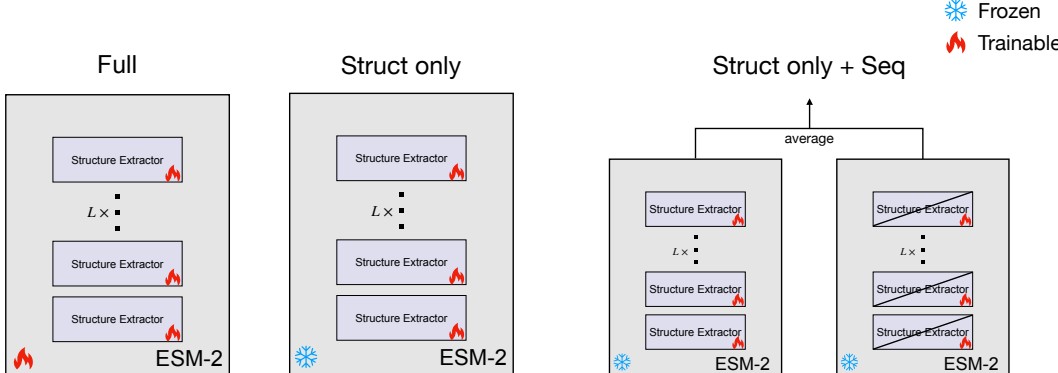

Figure 7: Illustration of different pretraining strategies: "Full" refers to the strategy where one updates the full PST model inclduing both ESM-2 and structure extractor weights. "Struct Only" refers to the strategy where only the structure extractor weights are being updated during training. "Struct Only" employs the same pre-training strategy as "Struct Only", and the only difference lies in the inference. Specifically, by bypassing the the structure extractors, the PST model trained with the "Struct Only" strategy is capable to obtain the base ESM-2's sequence representations. By averaging both structure and sequence representations, we obtain the "Struct Only + Seq" strategy.

