# OpenReview forum: "Endowing Protein Language Models with Structural Knowledge"
_ICLR.cc/2024/Conference — Submitted to ICLR 2024_

### Official Review · Reviewer_XuQU · 2023-10-23

**Soundness:** 3 good
**Presentation:** 3 good
**Contribution:** 2 fair
**Rating:** 6
**Confidence:** 5

**Summary:**

This paper introduces a novel approach to incorporate structural information into the protein representation learning process. By converting protein structural information into graph structures, the model employs a Graph Neural Network (GNN) to process the structural graph information of proteins. It then integrates the structural representations learned by the GNN with the representations of the ESM-2 model, ultimately resulting in a protein representation that has been enriched with structural information. The paper validates the model's performance on various protein function prediction tasks and compares the impact of different factors such as model size, pre-training strategies, and the amount of structural information on the model's effectiveness.

**Strengths:**

+ This paper introduces a protein representation learning method that integrates protein structural information with sequence-based models, which has a positive impact on the development of protein representation learning models.
+ The model proposes the direct integration of protein structural information into the pre-trained ESM-2 model, reducing the training cost and resource requirements.
+ The model's performance is validated across various model sizes, revealing insights into the impact of scaling on such models.

**Weaknesses:**

+ The main reported results in the paper (Table 1), where the model's performance is compared to ESM-Gearnet MVC, show only a slight advantage, and this advantage is likely due to the use of a stronger backbone model (ESM-2 vs. ESM-1b). Therefore, the results may not be very persuasive in demonstrating a good performance improvement for the PST model.
+ The approach of incorporating protein structural information into the ESM model using a GNN has been previously proposed in other papers[1]. While this paper just applies this approach to the field of protein representation learning, it lacks novelty, and it does not discuss the differences between these methods.
+ Table 1 does not report the performance of ESM-2 under end-to-end training conditions, and it does not specify important hyperparameters of the compared models, such as model size, which is a crucial factor affecting relative model performance.

[1] Zheng, Zaixiang, et al. "Structure-informed Language Models Are Protein Designers." (2023).

**Questions:**

+ Has the paper explored the impact of different backbone models on performance? For example, ESM-1b vs. ESM-2.
+ Since there are pre-trained GNN models designed for protein structural graph, such as GearNet[1], did the paper investigate the performance impact of using such pre-trained GNN models?

[1] Zhang, Zuobai, et al. "Protein Representation Learning by Geometric Structure Pretraining." The Eleventh International Conference on Learning Representations. 2022.

---

> ### Author Response · Authors · 2023-11-20
> **Author response to reviewer XuQU**
>
> > The main reported results in the paper (Table 1), where the model's performance is compared to ESM-Gearnet MVC, show only a slight advantage, and this advantage is likely due to the use of a stronger backbone model (ESM-2 vs. ESM-1b). Therefore, the results may not be very persuasive in demonstrating a good performance improvement for the PST model.
>
> In our new results, detailed under point 3 in the general comments, we have observed that PST models with fixed representations outperform all fine-tuned models, including ESM-Gearnet MVC, across all function prediction tasks. This finding combined with the enhanced practicality in terms of reduced computational resource requirements, clearly underscores the advantage of PST over fine-tuned models like ESM-Gearnet-MVC.
>
> Moreover, based on the results in a very recent preprint (https://arxiv.org/pdf/2303.06275.pdf), that we cite in our revised manuscript, we observe that the performances of ESM-Gearnet-MVC based on ESM-1b are very similar or even better than the one obtained using ESM-2, on most of the considered tasks.
>
> > The approach of incorporating protein structural information into the ESM model using a GNN has been previously proposed in other papers[1]. While this paper just applies this approach to the field of protein representation learning, it lacks novelty, and it does not discuss the differences between these methods.
>
> First, it is worth highlighting that our proposed architecture is significantly different from LM-Design, as elaborated in point 2.i of the general comment. Second, the central objective of LM-Design centers around protein design such as inverse folding, whereas our work primarily aims to enhance protein representations specifically for various tasks related to protein function prediction.
>
> > Table 1 does not report the performance of ESM-2 under end-to-end training conditions, and it does not specify important hyperparameters of the compared models, such as model size, which is a crucial factor affecting relative model performance.
>
> We have added the performance of ESM-2 under end-to-end training, and we notice that its performance is inferior compared to the PST (finetuned).
> Moreover, we have added the number of parameters of the ESM-2 model and of the corresponding PST model, as well as the ones of the baselines we run ourselves, in the appendix B.4. The number of parameters of the other models are not reported in the Gearnet paper.
>
> > Has the paper explored the impact of different backbone models on performance? For example, ESM-1b vs. ESM-2.
>
> We appreciate the reviewer's suggestion to explore the impact of various backbone models. Nonetheless, our current focus is on the state-of-the-art PLM, namely ESM-2, as it presents more relevance and potential for advancement compared to its predecessor, ESM-1b.
> Moreover, our investigations into the impact of model size have indicated that the quality of the base sequence model is directly proportional to the performance of the corresponding PST in general. Hence, it is a reasonable inference that a PST model built upon ESM-2 would likely surpass one based on ESM-1b.
>
> > Since there are pre-trained GNN models designed for protein structural graph, such as GearNet[1], did the paper investigate the performance impact of using such pre-trained GNN models?
>
> We would like to point out that in contrast to previous approaches that use a deep GNN to encode the structural information and add the encoded features atop a PLM, PST adopts a different approach. Specifically, we use a different two-layer GIN model (as described in section 4.1.2) at __each self-attention layer__ within the base ESM-2 model. Therefore, the weights learned in each GIN model should differ from one layer to another, making the application of a deep, pre-trained GNN model inapplicable in this context. We have clarified this by adding a brief introduction of our PST method at the beginning of Section 3.2 in the revised manuscript. See also our general comments above.

---

> > ### Comment · Reviewer_XuQU · 2023-11-22
> >
> > Thank you for your patient response. Your answers have helped clarify some issues, and I have reassessed the paper's score.

---

> > > ### Author Response · Authors · 2023-11-22
> > > **Reply to reviewer XuQU**
> > >
> > > Dear reviewer XuQU,
> > >
> > > We appreciate your recognition of our efforts to address the concerns raised in your review. It is encouraging to know that our rebuttal has successfully addressed your concerns. Should there be any additional questions or issues, we welcome you to raise them. We are eager to engage in further discussion to clarify any points and ensure the quality of our work meets the highest standards.

---

### Official Review · Reviewer_hmnV · 2023-11-01

**Soundness:** 2 fair
**Presentation:** 3 good
**Contribution:** 2 fair
**Rating:** 5
**Confidence:** 4

**Summary:**

The paper presents a novel framework, the Protein Structure Transformer (PST), aimed at enhancing the efficiency of protein language models (PLMs) by seamlessly integrating structural information derived from protein structures. Building on the foundation of the ESM-2 model, the PST refines the self-attention mechanism using structure extractor modules and leverages recent advances in graph transformers.

The model can be further pretrained on databases like AlphaFoldDB, enhancing its performance. An observation in the experiment is that improvements can be achieved by finetuning just the structure extractors, addressing concerns about parameter efficiency.

The PST demonstrates superior performance over ESM-2 in various prediction tasks related to protein function and structure.

**Strengths:**

1. The authors combine advances in graph transformers with existing protein language models to enhance the performance of protein function prediction tasks, showcasing a fusion of existing techniques to create something novel.

2. The empirical findings presented are comprehensive and demonstrate the PST’s performances on the EC, GO and some tasks from the ProteinShake benchmark, and specially parameter efficiency in training compared to the existing methods.

**Weaknesses:**

1. While the integration of a structure extractor with a PLM is presented as a unique proposition for function prediction tasks, it exits parallel studies in the protein representation field. For instance, LM-Design [1] proposes a similar structure adapter into PLMs that endows PLMs with structural awareness, and the structure adapter could access an arbitrary additional structure encoder (GNNs, ProteinMPNN etc.). As a result, this might raise concerns about novelty in comparison to existing methods.

2. A limitation in the study is the lack of an empirical exploration regarding the selection of different structure encoder. While the paper presents results based on a specific structure extractor module, it would have been insightful to see comparisons or benchmarks against various other structure encoding methodologies.

[1] Structure-informed Language Models Are Protein Designers, ICML 2023.

**Questions:**

1. The paper primarily focuses on the integration of a structure extractor with the ESM-2 model. However, with the availability of larger models like ESM2-3b, have there been empirical studies to assess the impact and advantages of the structure extractor? Specifically, as the scale of the PLM increases, does the benefit of adding a structure extractor diminish or remain consistent? It would be crucial to understand the interplay between model size and the structure extractor's efficacy.

---

> ### Author Response · Authors · 2023-11-20
> **Author response to reviewer hmnV**
>
> > While the integration of a structure extractor with a PLM is presented as a unique proposition for function prediction tasks, it exits parallel studies in the protein representation field. For instance, LM-Design [1] proposes a similar structure adapter into PLMs that endows PLMs with structural awareness, and the structure adapter could access an arbitrary additional structure encoder (GNNs, ProteinMPNN etc.). As a result, this might raise concerns about novelty in comparison to existing methods.
>
> First, it is worth highlighting that our proposed architecture is significantly different from LM-Design, as elaborated in point 2.i of the general comments. Second, the central objective of LM-Design centers around protein design such as inverse folding, whereas our work primarily aims to enhance protein representations specifically for various tasks related to protein function prediction.
>
> > A limitation in the study is the lack of an empirical exploration regarding the selection of different structure encoder. While the paper presents results based on a specific structure extractor module, it would have been insightful to see comparisons or benchmarks against various other structure encoding methodologies.
>
> Thank you for the suggestion. Our research draws significant inspiration from Chen et al. 2022 [1], leading us to choose GIN as our structural extractor due to its balance of performance and efficiency. Additionally, we believe that the quantity of structural information used in the structural extractor is more influential than the specific type of structural extractor chosen. We thus explored the impact of varying amounts of structural information in our study, as demonstrated in Section 4.4 and Figure 2.
>
> > The paper primarily focuses on the integration of a structure extractor with the ESM-2 model. However, with the availability of larger models like ESM2-3b, have there been empirical studies to assess the impact and advantages of the structure extractor? Specifically, as the scale of the PLM increases, does the benefit of adding a structure extractor diminish or remain consistent? It would be crucial to understand the interplay between model size and the structure extractor's efficacy.
>
> Our work includes scalability studies for various model sizes that can be accommodated by most GPUs, as depicted in Figures 2, 3, and 4. These studies indicate that the benefits of utilizing structural extractors are maintained across different scales. Although the advantages might diminish slightly for very large models, the utilization of structural information is still expected to offer benefits.
> Regarding extremely large models like ESM2-3b, we were only able to perform inference with it, as fitting a PST model of this size exceeded our capabilities. Interestingly, in the Variant Effect Prediction (VEP) task, our results show that ESM2-3b is outperformed by smaller versions of ESM2 models. For instance, ESM2-650M and the corresponding PST achieved a higher mean Spearman's correlation (0.489 and 0.501 respectively) compared to ESM2-3b's 0.456, highlighting the effectiveness of smaller models in certain applications.
>
> _Reference_
>
> [1] Dexiong Chen, Leslie O’Bray, and Karsten Borgwardt. Structure-aware transformer for graph representation learning. ICML 2022.

---

> > ### Author Response · Authors · 2023-11-22
> > **Kind reminder to reviewer hmnV for open discussion**
> >
> > Dear reviewer hmnV,
> >
> > Thank you for your thorough review of our work and the insightful feedback provided. We have carefully considered and responded to each of your comments in our rebuttal. With the deadline for author-reviewer discussion nearing, we respectfully request your review of our revisions and welcome any additional feedback or questions. Your further guidance is essential for the refinement of our work. We look forward to your response and extend our gratitude once more for your significant role in this process.

---

### Official Review · Reviewer_GUNq · 2023-11-01

**Soundness:** 3 good
**Presentation:** 3 good
**Contribution:** 3 good
**Rating:** 6
**Confidence:** 3

**Summary:**

This paper extends a pre-trained protein language model, ESM-2, with information from predicted structures. Architecturally, PST extends ESM-2 by adding a GNN module to the Transformer. PST also includes additional pre-training steps based on predicted structures from AlphaFold. When adapted to a range of downstream tasks, PST outperforms ESM-2.

**Strengths:**

* The paper is clearly written.
* The experiments attempt to offer careful comparisons with a strong model from prior work, ESM-2. PST outperforms ESM-2 on a range of protein tasks. An improvement can be observed even when only tuning the structure-related parameters.
* The paper offers strong evidence that structural information from predicted structures can offer improved performance. The proposed methods for incorporating structural information seem reasonable.

**Weaknesses:**

* A controlled comparison between ESM-2 and PST might have considered continuing MLM pre-training for ESM-2 for an equivalent number of steps as PST pre-training. However, the experiments showing that tuning only the structure-related parameters (section 4.4) is sufficient to improve performance partially address this.
* While the improvement from PST over ESM-2 is largest for smaller model sizes, the efficiency improvement is partially offset by the need to predict a structure.

**Questions:**

* nit: The BERT citation is not formatted correctly.

---

> ### Author Response · Authors · 2023-11-20
> **Author response to reviewer GUNq**
>
> > A controlled comparison between ESM-2 and PST might have considered continuing MLM pre-training for ESM-2 for an equivalent number of steps as PST pre-training. However, the experiments showing that tuning only the structure-related parameters (section 4.4) is sufficient to improve performance partially address this.
>
> The ESM-2 models have been pre-trained on the Uniref-50, a dataset much larger than the structure dataset (i.e. Swissprot) used for pre-training the PST models. Consequently, further MLM pre-training of ESM-2 on Swissprot could potentially lead to overfitting, negatively affecting model's performance in downstream tasks. Moreover, we believe that "updating only the structure-related parameters is sufficient to improve performance" should be interpreted as a strength of our method, rather than a weakness, as elaborated in the point 2.iv in the general comments.
>
> > While the improvement from PST over ESM-2 is largest for smaller model sizes, the efficiency improvement is partially offset by the need to predict a structure.
>
> While we acknowledge this is a good point, it is worth noting that structures can generally be sourced from precomputed databases (such as ESMFold and AlphaFoldDB), and are reusable for various tasks. Therefore, we argue that the additional computational costs due to structure predictions are quickly compensated by the reduced costs due to the data- and parameter efficiency of our models.
> Moreover, the effectiveness of our model based on fixed embeddings, without any additional finetuning, significantly reduces computational demands compared to fine-tuned models. This further strengthens our argument regarding the cost-effectiveness of our approach.
>
> > nit: The BERT citation is not formatted correctly.
>
> Thank you for noticing this point, we have addressed the formatting issue in the revised manuscript.

---

> > ### Comment · Reviewer_GUNq · 2023-11-21
> >
> > Thank for you for your reply. I confirm my original score of leaning towards acceptance.
> >
> > > we believe that "updating only the structure-related parameters is sufficient to improve performance" should be interpreted as a strength of our method
> >
> > Yes, this was the intended interpretation of my comment.

---

> > > ### Author Response · Authors · 2023-11-22
> > > **Reply to reviewer GUNq**
> > >
> > > Dear reviewer GUNq,
> > >
> > > Thank you for acknowledging our efforts in addressing the concerns raised in your review. It is encouraging to know that our rebuttal has successfully addressed your concerns. If there are any remaining questions or issues, please feel free to bring them to our attention. We are eager to engage in further discussion to clarify any points and ensure the quality of our work meets the highest standards.

---

### Official Review · Reviewer_daTw · 2023-11-02

**Soundness:** 3 good
**Presentation:** 3 good
**Contribution:** 2 fair
**Rating:** 5
**Confidence:** 5

**Summary:**

The paper introduces a novel framework called Protein Structure Transformer (PST) that enhances transformer protein language models specifically on protein structures, resulting in superior performance on several benchmark datasets. The PST integrates structural information with structure extractor modules, which are trained to extract structural features from protein sequences. The authors evaluate the performance of the PST on several benchmark datasets and compare it to the ESM-2 model. The results show that the PST outperforms the ESM-2 model on several tasks, including protein secondary structure prediction, contact prediction, and remote homology detection.

**Strengths:**

- The paper introduces a novel framework that enhances transformer protein language models specifically on protein structures, which is an important area of research in bioinformatics.
- The PST outperforms the ESM-2 model on several benchmark datasets, demonstrating the effectiveness of the proposed framework.
- The paper provides a detailed description of the PST framework and the structure extractor modules, which could be useful for researchers interested in developing similar models.

**Weaknesses:**

- Overall speaking, the novelty of this paper is enough. There are plenty of works that integrated the graph representation in protein representations, even alphafold. The authors mentioned about the minimal training cost and parameter efficient, while I acknowledge the good point. This is hard to be a strong strength.
- The framework actually is a GIN + ESM model, with the ESM freezed and GIN and head tuned. In Figure 1, however, the GIN is not clearly  presented. If GNN in the figure is the extractor, it means the GNN would be processed multiple L layers, this is somehow not necessary or why for this design?
- Small question, the GIN is randomly initialized, what's the difference between zero-initialized as you mentioned in the paper? Besides, is the node embedding in GIN is initialized by ESM token embedding?
- The paper does not provide a detailed analysis of the limitations of the proposed framework or potential areas for improvement.

**Questions:**

See above.

---

> ### Author Response · Authors · 2023-11-20
> **Author response to reviewer daTw**
>
> > Overall speaking, the novelty of this paper is enough. There are plenty of works that integrated the graph representation in protein representations, even alphafold. The authors mentioned about the minimal training cost and parameter efficient, while I acknowledge the good point. This is hard to be a strong strength.
>
> Please see the point "Novelty and impact of our work" in the general comments.
>
> > The framework actually is a GIN + ESM model, with the ESM freezed and GIN and head tuned. In Figure 1, however, the GIN is not clearly presented. If GNN in the figure is the extractor, it means the GNN would be processed multiple L layers, this is somehow not necessary or why for this design?
>
> Note that in the transformer architecture, the node embeddings used for each GNN corresponds to the residue embeddings of the respective layer, and these embeddings can vary from one layer to another. Consequently, the query, key, and value matrices have to be re-computed at each layer. Please see also our first point of the general comments for a clarification of the PST architecture.
>
> > Small question, the GIN is randomly initialized, what's the difference between zero-initialized as you mentioned in the paper? Besides, is the node embedding in GIN is initialized by ESM token embedding?
>
> See __Clarification of the PST architecture__ in our general response, where we clarified our general approach, and should address general questions about the methodology. Now to answer the reviewer's questions: i) the weights in GIN, i.e., $\theta$ is randomly initialized while $W_s$ is initialized to zero such that the _PST model without training_ makes the same prediction as the base ESM-2 model. ii) The node embedding in the GIN before each self-attention layer is given by the residue embedding at the respective layer. We have added a clarification of PST in Section 3.2 of the revised version.
>
> > The paper does not provide a detailed analysis of the limitations of the proposed framework or potential areas for improvement.
>
> We kindly direct the reviewer's attention to the limitations discussed in section 4.4 and Appendix A.

---

> > ### Author Response · Authors · 2023-11-22
> > **Kind reminder to reviewer daTw for open discussion**
> >
> > Dear reviewer daTw,
> >
> > Thank you for your thorough review of our work and the insightful feedback provided. We have carefully considered and responded to each of your comments in our rebuttal. With the deadline for author-reviewer discussion nearing, we respectfully request your review of our revisions and welcome any additional feedback or questions. Your further guidance is essential for the refinement of our work. We look forward to your response and extend our gratitude once more for your significant role in this process.

---

### Official Review · Reviewer_wJCU · 2023-11-10

**Soundness:** 2 fair
**Presentation:** 2 fair
**Contribution:** 2 fair
**Rating:** 5
**Confidence:** 5

**Summary:**

This paper proposes the Protein Structure Transformer (PST) built on the protein language models (PLM), such as ESM-2, to incorporate structural information into PLM for the purpose of obtaining structural-aware protein representations. It takes the 2D-ordered protein graph as input and devises the structure extractor modules within self-attention architecture. The experiments show its superiority compared to the vanilla PLMs.

**Strengths:**

1. This paper presents the simple yet effective structure extractor modules to inject structural knowledge into the vanilla sequence-based PLMs, thus enhancing the representation ability of PLMs.
2. The improved performance on several downstream tasks is promising. Especially, the parameter-efficiency Sructrual Only models can match the full model on a variety of tasks, which indicates the PST can serve as the flexible plug-in modules to any PLMs to enhance their representation ability.

**Weaknesses:**

1. The paper writing style needs to be further improved. The Method spends a lot of pages introducing the principle of ESM-2, which is not the contribution of the paper. On the contrary, the introduction of the PST framework is vague. How the node embedding output by GNN is further incorporated into the residue embedding of the original ESM? Did they just add, concentrate? or viewed as the individual token embedding which is conducted with self-attention to the residue embedding of the original ESM? I highly suggest the authors to add more details.
2. The introduction of structural knowledge of 3D protein structures is limited. This paper just compressed the 3D structures into 2D graphs without considering the 3D geometric features, such as the SE(3)-Equivariant features, which have already been confirmed to be critical in modeling the 3d protein structures or the protein-protein docking patterns by many works, such as Alphafold2, EquiDock, etc. The only considered feature is the distances between nodes severed as the edge attributes while performing less-satisfied on the downstream tasks.
3. The paper only adopts the subset of AlphaFoldDB. I wonder if the author ever tried other structural databases such as metagenomic protein databases predicted by ESMFold? How do the data quality and quantity affect the performance of PST?
4. The downstream tasks adopted in this work are limited. As the protein structure information decides the physical and chemical properties of proteins, the downstream tasks should not only be performed on protein structure prediction tasks, but also on other tasks, like protein solubility prediction,  secondary structural prediction, etc. Like xTrimoPGLM (chen et al. ) and Ankh (Elnaggar, et al) do.
5. Besides the ablation studies claimed in the pre-training strategy, the author should add Seq-only to get rid of the reason solely brought by the sequence-based training.
6. "While PST typically surpasses its base ESM-2 counterpart at similar model sizes, this performance gain tapers off with increasing model size. "Does this observation indicate that if we adopt a huge amount of protein sequence data and also the large scale of PLMs,   the structural knowledge can be well-captured on the sequence-based pertaining, like xTrimoPGLM with 100B dominates on most of the downstream tasks?
7. From the Table.1, although the PST with fixed representation outperforms the GearNet MVC, the end-to-end PST(fintuned) lies behind the GearNet MVC on the fold classification tasks. How to explain this?



Ganea, et al. "Independent se (3)-equivariant models for end-to-end rigid protein docking." arXiv preprint arXiv:2111.07786 (2021).

Chen, et al. "xTrimoPGLM: unified 100B-scale pre-trained transformer for deciphering the language of protein." bioRxiv (2023): 2023-07.

Elnaggar, et al. "Ankh☥: Optimized protein language model unlocks general-purpose modeling." bioRxiv (2023): 2023-01.

**Questions:**

See weakness.

---

> ### Author Response · Authors · 2023-11-20
> **Author response to reviewer wJCU**
>
> >1. The paper writing style needs to be further improved.
>
> We thank the reviewer for the suggestion and have updated the manuscript accordingly (please see __Clarification of the PST architecture__ in our general comments). As a result, a thorough description of the ESM-2 model is crucial as it lays the groundwork for the mechanism discussed above.
>
> >2. The introduction of structural knowledge of 3D protein structures is limited.
>
> We acknowledge the reviewer's point about the importance of using additional geometric information in several structure prediction tasks. However, its impact on protein function prediction, the primary focus of our work, remains less evident. To elucidate how the incorporation of different amounts of geometric information impacts PST models, we direct the reviewer's attention to our ablation study presented in Section 4.2 under "Amount of structural information required for refining ESM-2". Our results suggest that while incorporating more structural features, such as the distance information used in PiFold [1], improves training accuracy, it negatively affects the model's performance in downstream tasks, as shown in Figure 2.
>
> > 3. The paper only adopts the subset of AlphaFoldDB.
>
> We did not train our PST model on ESMFold, which is out of the computational capacity in our group. However, our results indicate that fine-tuning of ESM-2 with structural data, even on a relatively modest-sized structure dataset such as Swissprot (~550k structures) can already enhance the quality of protein representations. We are eager to explore the full potential of PST on larger datasets like ESMFold in future work.
>
> > 4. The downstream tasks adopted in this work are limited.
>
> - First, our method was evaluated on a wide range of tasks, such as enzyme commission number prediction and GO term prediction given a protein as used to evaluate Gearnet and DeepFRI. To extend our analysis, we use the novel ProteinShake tasks to evaluate the quality of the embeddings on other biologically meaningful tasks such as binding site prediction, structural class prediction and protein family prediction. We believe that these tasks collectively provide a thorough overview of a wide range of protein-related applications.
> - Second, our focus was primarily on protein function prediction task, which is a common focus of many representation learning approaches such as Gearnet and DeepFRI, allowing us to benchmark our representations against existing approaches.
> - Third, we found that the suggested benchmarks did not contain UniProt IDs which prevented us from retrieving the exact corresponding structures from AlphaFoldDB needed for direct comparison on the Ankh datasets. Instead we were able to apply the Ankh model to our datasets for the EC and GO tasks (see updated Table 1). Regarding xTrimPGLM, we could not easily find the model weights online.
>
> > 5. Besides the ablation studies claimed in the pre-training strategy, the author should add Seq-only to get rid of the reason solely brought by the sequence-based training.
>
> First, we want to clarify that in the "pretraining strategies" section, the "Struct Only" strategy specifically consists of updating __only the structure extractors__ in the PST model. In this strategy, the weights of the base ESM-2 model remain unchanged during training, allowing the trained PST model to also generate the sequence representations (which is the same as the base ESM-2's representations) at inference, by bypassing the structure extractors in the trained PST. The "Struct Only + Seq" strategy simply takes the average of both structure and sequence representations. This was fully detailed in our submitted version. We have further included Figure 6 to illustrate the difference between these strategies in the revised manuscript.
>
> As for the "sequence-only training" referenced by the reviewer, we believe this corresponds to the ESM-2 model. We encourage the reviewer to refer to Section 4.3 of our paper, where we present an extensive comparison between PST and ESM-2.

---

> > ### Author Response · Authors · 2023-11-20
> > **Author response to reviewer wJCU (cont.)**
> >
> > > 6. "While PST typically surpasses its base ESM-2 counterpart at similar model sizes, this performance gain tapers off with increasing model size. "Does this observation indicate that if we adopt a huge amount of protein sequence data and also the large scale of PLMs, the structural knowledge can be well-captured on the sequence-based pertaining, like xTrimoPGLM with 100B dominates on most of the downstream tasks?
> >
> > - First, this observation mentioned does not imply that "the largest models always perform the best in all tasks". For instance, in tasks such as structural class and protein family prediction (shown in Figure 3), models incorporating structural knowledge can outperform even very large sequence-based models. Therefore, the benefit of incorporating structural knowledge remains significant.
> > - Second, We acknowledge the relevance of huge protein language models like xTrimoPGLM, and have added a discussion in Section 2. While scaling up sequence models might seem advantageous, it also brings practical challenges, notably in terms of data and computational requirements. In contrast, structure-focused models like PST are more efficient in both data usage (e.g. PST was only trained on Swissprot) and computational resources. Within modest GPU memory constraints, PST using fixed representations generally surpasses the performance of state-of-the-art sequence models like ESM-2.
> >
> > > 7. From the Table.1, although the PST with fixed representation outperforms the GearNet MVC, the end-to-end PST(fintuned) lies behind the GearNet MVC on the fold classification tasks. How to explain this?
> >
> > First, we would like to emphasize that PST (finetuned) achieves performances at least comparable to GearNet MVC (finetuned) on the fold classification at both the superfamily and family levels. Moreover, please note that for this dataset, each class typically has a limited number of samples, only having around 10 per class on average. This data scarcity renders the fine-tuning particularly challenging. GearNet MVC conducted extensive hyper-parameter tuning, while PST did not involve any hyperparameter optimization for fine-tuning. Finally, it is worth highlighting that PST significantly outperforms GearNet on function prediction, which is the main focus of this work, as elaborated in the second point of the general comments.
> >
> > > Concerns for ethics: Yes, Potentially harmful insights, methodologies and applications
> >
> > Although we recognize that advanced pre-trained models have the potential to be misused for harmful purposes (as discussed in Appendix A), we do not anticipate any direct ethical concerns stemming from the application of our research.
> >
> > _Reference_
> >
> > [1]: Gao et al. PiFold: Toward effective and efficient protein inverse folding. ICLR, 2023.

---

> > > ### Comment · Reviewer_wJCU · 2023-11-22
> > >
> > > Thanks for the authors' response. It addresses several of my concerns. Thus I improved the paper's score.

---

> > > > ### Author Response · Authors · 2023-11-22
> > > > **Reply to reviewer wJCU**
> > > >
> > > > Dear reviewer wJCU,
> > > >
> > > > We appreciate your recognition of our efforts to address the concerns raised in your review. It is encouraging to know that our rebuttal has successfully addressed your concerns. Should there be any additional questions or issues, we welcome you to raise them. We are eager to engage in further discussion to clarify any points and ensure the quality of our work meets the highest standards.

---

### Author Response · Authors · 2023-11-20
**General comments**

Dear reviewers,

We are grateful for your dedication in offering detailed and insightful feedback, which we believe has significantly strengthened our paper. We would like to express our appreciation for your positive remarks:

- The paper offers strong evidence that structural information from predicted structures can offer improved performance on protein function prediction tasks.
- The paper presents a simple yet effective parameter efficient structure extractor module to inject structural knowledge into any vanilla sequence-based PLMs, thus enhancing the representation ability of PLMs.
- The PST outperforms the ESM-2 model on several benchmark datasets, demonstrating the effectiveness of the proposed framework.
- The paper is clearly written.
- The model's performance is validated across various model sizes, revealing insights into the impact of scaling on such models.

While we will address specific comments in the individual reviews (and in the revised PDF, where the changes can be seen in red), we want to address here three general points raised by several reviewers:

1. __Clarification of the PST architecture__
We have updated Section 3.2 and Figure 1 to more clearly explain our methodology. Specifically, at each layer, as outlined in Equation (5), the GNN's node embeddings $\varphi_{\theta}(X,G)$ are used to update the standard query, key and value matrices provided by the base ESM-2. This procedure makes all self-attention layers in the ESM-2 model structure-aware.
__It is important to highlight that this procedure is applied across all self-attention layers within the base model__, making PST a unique framework for transformer-based PLMs and differentiating it from all previous protein structure models.

2. __Novelty and impact of our work:__
    - i) The architecture of our PST model differs significantly from previous approaches, such as LM-Design and ESM-GearNet. Instead of merely adding a structure-aware adapter on top of a PLM model (typically a transformer) such LM-Design and ESM-GearNet, our approach incorporates structural information directly into each self-attention layer of the PLM. This approach allows for a deeper interaction between structural and sequence features, setting our model apart from previous models. We have included a discussion on this difference in Section 2 of the revised version.
    - ii) A pivotal aspect of our work is demonstrating the effectiveness of incorporating structural knowledge in sequence models for protein function prediction. Our experiments demonstrate that PST outperforms the strong baseline ESM-2 in these tasks.
    - iii) Our new results (see point 3 below) reveal a significant practical advancement: our PST model with fixed representations and a finetuned MLP classification head substantially exceeds the performance of all fully fine-tuned models, including the ESM-GearNet MVC. This enhancement in PST not only showcases its superior efficacy but also increases its practical applicability, especially compared to models that depend extensively on heavy fine-tuning.
    - iv) We are the first to show that partial fine-tuning of ESM-2 with structural data, even on a relatively modest-sized structure dataset (SwissProt structures from AlphaFoldDB, ~550k), can substantially enhance the quality of protein representations. Therefore, this approach makes a significant step forward in protein representation learning.

---

> ### Author Response · Authors · 2023-11-20
> **General comments (cont.)**
>
> 3. __Performance compared to finetuned models:__ In our submitted version, we opted for simplicity and efficiency by reducing the representation dimensions to 2048 using PCA, and did not perform any hyper-parameter tuning on the downstream tasks. This did not lead to a fair comparison with ESM-Gearnet which exhibited extensive hyper-parameter tuning. Therefore, our revised version eliminates the dimension reduction, using the full-scale representations instead. Additionally, we have incorporated dropout layers in the MLP classification head to prevent overfitting. New results are updated in Table 1 of the revised manuscript as well as in the following table (with Fmax/AUPR as the metrics). Our new findings underscore that PST with fixed representations surpasses all fully fine-tuned models in terms of AUPR, notably in the GO-BP and GO-CC task. This outcome holds considerable importance for two reasons: i) it contrasts with trends in NLP and computer vision, where fine-tuning typically prevails over fixed representation models; ii) The enhanced practicality of PST with fixed representations, especially in terms of reduced computational resource requirements, signifies a clear contribution in the field.
>
> | Method                       |     EC      |    GO-BP    |    GO-MF    |    GO-CC    |
> | ---------------------------- |:-----------:|:-----------:|:-----------:|:-----------:|
> | ESM-Gearnet MVC (finetune)   | 0.894/0.907 | 0.516/0.301 | 0.684/0.621 | 0.506/0.359 |
> | PST (finetuned)              | 0.901/0.922 | 0.493/0.348 | 0.688/0.655 | 0.509/0.387 |
> | Gearnet MVC                  | 0.826/0.852 | 0.428/0.303 | 0.594/0.589 | 0.433/0.337 |
> | ESM2 (fixed representations) | 0.892/0.910 | 0.509/0.355 | 0.686/0.629 | 0.529/0.394 |
> | PST (fixed representations)  | 0.899/0.918 | 0.513/0.371 | 0.686/0.637 | 0.541/0.387 |

---

### Meta-Review · Area_Chair_f8tV · 2023-12-14

**Metareview:**

This work aims to study structurally-informed protein representations. It introduces the Protein Structure Transformer (PST) model. It enhances existing protein language models (sequence models, e.g., ESM-2) with structure information. It leverages a 2D-ordered protein graph as input and design a structure extractor with self-attention. Experiments on protein function prediction suggests interesting results, that is, outperformance over ESM-2.

The problem studied is of importance, as protein LMs have gained significant interest and produced promising results. Similar to LLMs, these protein LMs primarily depend on the sequence modeling capabilities of Transformers. The work focuses on exploring integrating protein structures into LMs.

However, at its current form, the proposed method and its empirical demonstration fail to impress the reviewers and the AC. The GIN + ESM combination does not bring substantial technical advances into this problem, and the experimental improvements (with new baselines added in Table 1) are marginal, rendering the overall evidence unconvincing. In addition, the organization and presentation of this work need refinement, as highlighted by the first reviewer.

**Justification For Why Not Higher Score:**

This submission receives borderline ratings (5, 5, 5, 6, 6) with three negative ones. The updated paper during rebuttal (new baseline results in Table 1) suggests the proposed technique does not strong performance compared to baselines. Overall, this work by leveraging existing techniques in a new manner does not excite the reviewers.

**Justification For Why Not Lower Score:**

n/a

---

### Decision · Program_Chairs · 2024-01-16

Reject